# Stakeholder considerations on acceptability and implementation of a novel rapid test for acute HIV infection: A qualitative study in Indiana

Natalia M. Rodriguez[1,2]*, Lara Balian[1], Ishita Kataki[1], Cealia Tolliver[1], Julio Rivera-De Jesus[2], Jacqueline C. Linnes[1,2]

**1** Department of Public Health, College of Health and Human Sciences, Purdue University, West Lafayette, Indiana, United States of America, **2** Weldon School of Biomedical Engineering, College of Engineering, Purdue University, West Lafayette, Indiana, United States of America

\* natalia@purdue.edu

## Abstract

Acute HIV infection (AHI) is the most infectious stage of HIV, yet existing rapid tests cannot reliably detect HIV in this early phase and require up to 90 days post-exposure for accurate results. Laboratory-based nucleic acid tests can detect AHI but are often inaccessible to high-risk populations with limited healthcare access. Novel rapid AHI tests are in development to bridge this gap by enabling earlier, decentralized detection. This study aims to explore the acceptability and future implementation considerations of such a test through engagement with clients (n = 5) and staff (n = 14) of Indiana-based HIV service organizations. Guided by human-centered design frameworks, semi-structured interviews examined experiences with HIV testing, acceptability of a proposed novel rapid AHI test that could detect infection about one month earlier but would require longer time to results (~60 minutes), and preferences for potential end-users (self, community health workers (CHW) or peers). Transcripts were thematic analyzed. Participants strongly supported earlier detection despite longer wait times, describing the trade-off as worthwhile for reducing transmission, initiating treatment sooner, and alleviating anxiety following high-risk exposures. Clients emphasized the benefits of convenience and peace of mind, particularly for people who use drugs or experience stigma. Staff highlighted the potential to retain individuals who are often lost to follow-up and to strengthen linkage to care. Concerns focused on organizational feasibility of accommodating longer test times and client expectations shaped by existing "rapid" tests. CHW/peer-led testing was preferred due to their ability to build trust, provide education and navigation, while self-testing was viewed as useful for overcoming barriers such as stigma and access, though concerns about accuracy, interpretation, and lack of counseling were noted. Stakeholder insights underscore the importance of accuracy, usability, affordability, and integration of counseling and care navigation in designing and implementing AHI rapid tests to maximize their impact among high-risk populations.

**Data availability statement:** Data contain potentially identifying or sensitive patient information and access is thus restricted by the Purdue University Institutional Review Board (irb@purdue.edu). Deidentified data, coding schemes, and interview guides are available by request from the corresponding author, Natalia Rodriguez, at natalia@purdue.edu.

**Funding:** This work was supported by funding from the National Institutes of Health (NIH) National Institute on Drug Abuse award DP2DA051910 (J.C.L.) and Gordon and Betty Moore Foundation (GBMF) award 9687 (J.C.L.). The funders had no role in study design, data collection and analysis, decision to publish, or preparation of the manuscript. The content is solely the responsibility of the authors and does not necessarily represent the official views of the NIH or GBMF.

**Competing interests:** We have read the journal's policy and the authors of this manuscript have the following competing interests: J.C.L. is an inventor on 3 issued US patents for the MicroRAAD (11,529,626, 11,628,434, 12,111,529) and is a co-founder and part owner of two startup companies in portable diagnostic and monitoring technologies: Rescue Biomedical LLC, and EverTrue LLC. These are documented in an annual financial conflict of interest report that is reviewed and approved by Purdue University's Office of Research for Ethics and Compliance. The other authors declare no competing interests.

## Introduction

The persistent burden of HIV (39 million worldwide [1] and 1.2 million in the U.S. [2]), particularly among marginalized populations, along with suboptimal testing rates, highlights the need to expand and innovate HIV testing efforts. HIV disproportionately affects sub-populations in the U.S., including people who inject drugs (PWID) [3]. Despite some success in reducing transmission among PWID through syringe service programs, the expanding injection drug epidemic and effects of COVID-related reduction in healthcare access have led to HIV outbreaks among PWID, who are already marginalized from the mainstream HIV care continuum [4,5]. The higher prevalence of HIV compounded with a range of social, economic, and demographic factors (e.g., stigma, education, income, rurality) can further increase risk for transmission, affect access to testing and care services, and potentially lead to worse outcomes and quality of life after HIV diagnosis in these high-risk populations [6].

HIV testing allows PLWHIV to make informed decisions about behaviors associated with HIV transmission, and once an individual receives a positive result they can then engage in HIV treatment and reduce viral load to untransmissible levels. Yet, almost 40% of new HIV infections in the U.S. are transmitted from people who did not know they were infected, and 13% (158,500) of people don't know their status [7]. While those who reported at least one high-risk behavior for HIV (e.g., injecting drugs, unprotected sex) had higher rates of testing than those who did not (65.2% compared to 44.2% in 2016), 34.8% still had never been tested, and 65.8% had not been tested in the past 12 months [8].

The highest risk of HIV transmission occurs in the acute HIV infection (AHI) phase, the first stage of infection that occurs in the first 2–4 weeks [9], yet current rapid tests fail to detect HIV in this stage. Existing tests that are most used to screen for HIV are rapid antibody or antibody/antigen tests that can produce results within 1–30 minutes from fingerstick blood or oral fluid. These tests require a waiting period of up to 90 days after exposure as they cannot detect the virus at the AHI stage when individuals are most likely to spread HIV to others. Novel rapid AHI tests, designed to detect infection earlier and closer to the time of exposure, could transform testing by identifying individuals who are most infectious and linking them to care sooner. Such tests are particularly relevant for high-risk populations, including people who inject drugs, who often face barriers to accessing traditional healthcare.

The FDA previously approved two point-of-care antigen/antibody tests designed to shorten the diagnostic window to less than one month [10]. Field tests of the Ag/Ab assays demonstrated improved detection of AHI compared to Ab only tests, however with highly variable performance ranging from detection of 28–64% of AHI infections [11–13]. While laboratory-based nucleic acid tests can detect HIV infection earlier (10–33 days after exposure), these require intravenous blood draw, expensive equipment, time-consuming protocols, cold-chain reagent storage and trained personnel. As such, researchers are working to further advance rapid HIV testing to detect viral RNA during the acute phase and at the point-of-care (POC) [14–19]. Dr. Linnes and colleagues are developing a POC AHI test capable of detecting the

equivalent of <1000 HIV virions/mL from a finger prick blood sample within 60 minutes. This test, called MicroRAAD, is a fully integrated, handheld, sample-to-answer device that uses reverse-transcription loop mediated isothermal amplification (RT-LAMP) to amplify HIV-1 RNA and provides a colorimetric readout for the amplification products via a lateral flow immunoassay. This platform can currently detect the virus within 90 minutes [15], and is being further developed with the goal of achieving results within 60 minutes. One of the most novel aspects of this device is achieving detection through whole blood sampling by automatically performing the multistep processes of isolating the virus from the sample, amplifying the RNA, and then transferring RT-LAMP products to the detection zone; making this a straightforward molecular testing platform for AHI. The test is designed to empower those at high-risk for HIV to detect and monitor their HIV status earlier following potential exposures.

This new testing paradigm has the potential to transform the HIV care continuum by making critical AHI detection possible at sites ranging from substance abuse treatment facilities to syringe services, addressing access barriers in areas of high HIV risk. However, without ensuring that a diagnostic POC technology is linking patients into the existing care continuum, even the most exquisitely sensitive POC devices will fail to make a difference in clinical outcomes. The success of this, and all technology platforms, will depend on the extent of adoption by, and delivery to, end-users. According to human-centered design frameworks, this requires examination of context-specific needs early and throughout the design process with end-users and experts who have lived experience or engage with the priority communities of an innovation [20].

The Ending the HIV Epidemic initiative in the U.S. aims to reduce new incidences of HIV by 90% by 2030 and to "diagnose all people living with HIV as early as possible" [21]. Marion County, Indiana represents one of the 48 priority counties that account for more than half of all new HIV diagnoses in the U.S [22]. This study sought to engage key stakeholders from organizations across Indiana who conduct HIV testing (direct users) with PWID and other high-risk populations, and clients of syringe services or HIV testing services (indirect users), to inform the development and implementation of the novel rapid AHI test. The objectives of this study were to 1) understand context-specific experiences with and barriers to HIV testing in Indiana; 2) assess the acceptability of this novel AHI test (that can detect the acute phase of HIV but takes longer to produce results than current HIV rapid tests) in this context, and 3) identify the target end-user and related implementation considerations for the test.

## Methods

### Ethics Statement

This study was approved by the Institutional Review Board of Purdue University (IRB-2020–685; IRB-2021–1437). All participants gave verbal informed consent.

### Study Design

This qualitative descriptive study employed human-centered design frameworks, in-depth interviews, and thematic analysis [23] in order to center participants' experiences and perspectives and facilitate contextually informed insights and solutions related to the development and implementation of a rapid AHI test. The Consolidated Criteria for Reporting Qualitative Research (COREQ) [24] was followed and reported in S1 File. The study team was comprised of faculty, staff and undergraduate and graduate student research assistants who completed the CITI program's human-subjects research course and training in qualitative research methods. There was no established relationship between the study team and participants prior to interviews.

Semi-structured interviews were conducted between September 2020 – December 2021 with staff of organizations that conduct HIV testing in Indiana and clients who receive services at HIV organizations and/or syringe service programs in Indiana.

Staff of organizations were recruited via snowball sampling. Organizations with publicly available email addresses were emailed about the study, instructed on how to participate if interested, and encouraged to pass along the opportunity to other colleagues and HIV service organizations. Any staff member of an HIV care organization, defined as any organization that provides HIV testing and linkage to care (e.g., clinic, health department, syringe service programs), was eligible to participate if they oversaw or conducted HIV testing (direct users of HIV tests).

Clients (indirect users of HIV tests) were recruited through passive methods, given the vulnerability of this population and the highly sensitive topic. Flyers were posted at local HIV organizations, with information about the study and researchers' contact information. Eligibility was limited to those 18 years and older and those who received services at syringe service programs or HIV testing organizations in Indiana. Due to COVID-19 restrictions during the first year of the study, all interviews conducted before October 2021 were conducted virtually via Zoom. After that, all participants were given the option to be interviewed in-person, over the phone, or via Zoom, as they preferred.

Two interview guides were developed and tailored to each participant group (staff and clients) based on human-centered design frameworks that call to understand unique perspectives, experiences, and context-specific factors of users and stakeholders [20], literature review, and investigator-initiated questions based on key device development considerations (S1 Table). To glean insights on the intended end-user for an AHI HIV rapid test, clients and staff were asked about their thoughts on self-testing, testing by a community health worker (CHW), defined as a frontline public health worker with a close understanding of the community served who acts as liaison between the community and health/social services [25], or a peer recovery coach, defined as a person who has been successful in recovery from addiction or mental health issues and assists others in their recovery process [26].

Clients and staff were informed about the proposed AHI test currently in development, which could detect HIV up to 1 month earlier than existing rapid tests but would take longer to run than current rapid tests (around 60 minutes). Participants were then asked, "Do you think this trade-off would be worthwhile to clients or providers?" and "Do you think there is a need to be able to detect HIV earlier for PWID and other at-risk populations?"

Interviews were conducted over phone or Zoom, a video communication platform, by staff and undergraduate and graduate student research assistants who completed the CITI program's human-subjects research course and training in qualitative research methods under the supervision of the co-principal investigators (NR, JL). There was no established relationship between the study team and participants prior to interviews. The interviews ran between 30–60 minutes and were audio-recorded and transcribed using Otter.ai, which were later reviewed for accuracy by members of the study team. A target number of interviews was not determined a priori but rather by recruitment capacity and during ongoing analysis toward thematic saturation, where each additional interview did not reveal new themes. Pseudonyms were used for all participants. Following a thematic analysis approach[CITE], codebook for each participant group was developed using inductive and deductive methods based on the interview guide, research questions, and preliminary transcript review and data familiarization (phase 1). Each interview was independently coded in NVivo [27], a qualitative coding software, by at least two independent coders on the study team trained in qualitative coding methods. The two coders met to reach consensus, and any coding discrepancies were brought to the larger study team to reach agreement (phase 2). Fully coded transcripts were analyzed to identify patterns and preliminary themes (phase 3) that were then reviewed and finalized as presented in the results section (phase 4–5).

## Results

A total of 19 interviews were conducted (n = 14 staff, direct users of HIV tests and n = 5 clients, indirect users of HIV tests). Of the 14 staff interviewed, five were directors or managers of Testing and Outreach units in their organization, while the remaining 10 directly conducted HIV and/or sexually transmitted infection (STI) testing. Most of the people who performed HIV testing also provided or coordinated counseling and outreach services, and two identified as a Peer Recovery Specialist or CHW. Years of experience in these or similar roles ranged from less than 1 year to over 20 years with an average

of 6.3 years. Of the five clients interviewed, two identified as male, two as female, and one unreported. Most identified as White (4) and one person as Hispanic. Two clients were experiencing homelessness, four were unemployed (no income reported), and education levels ranged from high school to completion of a master's degree. Almost all had used intravenous drugs (4). Those who injected drugs reported sharing needles never (2), rarely (1), and half the time (1). The number of sexual partners in the past year ranged from 1-3. All clients knew their current partner's HIV status and reported low condom use (3 never used condoms and 2 rarely used them).

The staff interviewed represented 10 organizations that provide HIV testing services across six counties throughout Indiana (Allen, Marion, Monroe, Scott, Tippecanoe, Vanderburgh), including county health departments, syringe service programs, and healthcare clinics. Some organizations were part of larger networks and partnerships, serving up to 45 counties and partnering with up to 75 other HIV/AIDS organizations. Most organizations also provided education and outreach on HIV and STIs, ranging from positing flyers in public spaces to actively going out to the community with testing kits, as well as healthcare navigation, and medical and non-medical case management (e.g., housing assistance). About half of the organizations also provided HIV treatment and PrEP services. While few organizations were specifically focused on PWID and had a syringe service or harm reduction program, all organizations had experience working with high-risk populations and many targeted specific efforts to further reach populations in their area at high-risk for HIV or who have low utilization of HIV testing services.

**Experiences and Barriers to HIV Testing**

Fig 1 summarizes the Indiana HIV testing experience as described by interviewees. From the organization perspective, a client may seek testing once because of a known or suspected exposure whereas others test regularly, from every 3 months to yearly, likely because of risky behaviors. Clients shared that choosing to get tested depended on their individual risk assessment such as drug use and sexual activity. Those who viewed their risk as high (e.g., due to injection drugs) tended to test more regularly and frequently. Clients reported accessing testing at treatment and recovery centers (e.g., needle exchange programs), HIV testing organizations, and/or doctor's offices and were last tested anywhere from 1-3 months prior to their interview, either by a rapid test or lab-based blood test. All staff reported using rapid HIV testing as

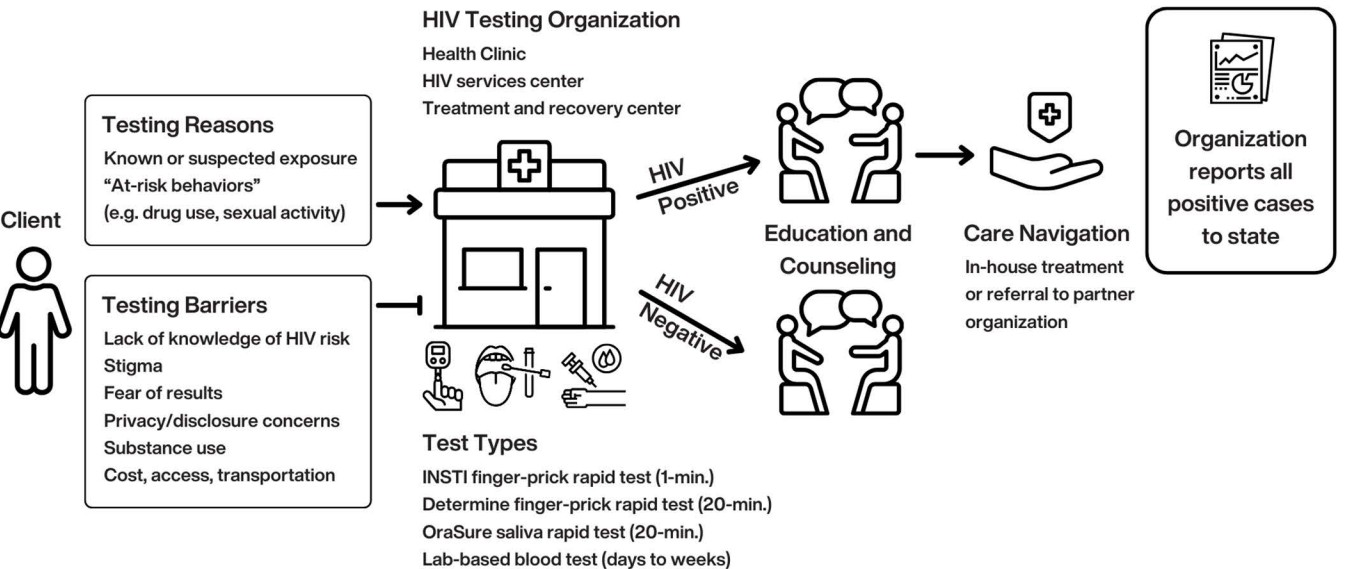

**Fig 1. HIV Testing Experience Pathway.**

the initial test for clients. INSTI® HIV-1/HIV-2 Antibody Test by bioLytical Laboratories Inc. [28] and Determine™ HIV-1/2 Ag/Ab by Abbott [29] were specifically mentioned as initial rapid tests that detect HIV from a finger prick of blood and yield results in one-minute and 20 minutes, respectively. OraQuick ADVANCE® [30], another rapid test that detects HIV antibodies, from oral fluid and yields results in 20 minutes, is used by staff as a second rapid test to confirm only positive INSTI results. Clients who test positive using a rapid test would then take a lab-based test that can take up to a week for results. The positive lab result officially confirms the HIV case so that the organization can report it to the Indiana State Health Department, per state law.

Clients who test positive are connected with a case manager to link directly to care within their organization, often starting them on treatment same day. Organizations who do not offer treatment services refer clients to other partner organizations. Many also noted that providing education before, during, and after testing is considered crucial for both positive and negative tests.

While most clients found HIV testing to be accessible, staff mentioned key access barriers including lack of awareness of where or how to access testing, concerns about cost, and transportation, specifically for high-risk populations such as PWID, people experiencing homelessness, or people who are incarcerated. Stigma was viewed as one of the most significant barriers to testing, for clients, in terms of drug use, fear of results, and worrying about other people learning their HIV status. From the staff perspective, stigma as a testing barrier is related to societal attitudes and beliefs about drug use and sex, and something they actively try to improve through education and counseling, *"I think stigma is a huge reason, obviously, that people stay away from getting tested. They don't want to feel dirty. They don't want to be judged for their sexual activities" (James, HIV/STI Tester).* Stigma was also associated with lack of knowledge about HIV and resulting misconceptions about HIV risk, or fear of getting tested because others might find out: *"they're afraid, in a small town, that somebody might find out that they're being tested or what if they are positive, then what are they going to do? People are just afraid, a lot of times to find out (Bonnie, HIV Tester/Counselor).*

## Acceptability of AHI Rapid Test

Acceptability of the described AHI rapid test was generally high among both clients and staff, though their perspectives emphasized different aspects of its value. Across groups, participants agreed that there is an urgent need to detect HIV earlier in high-risk populations, and most felt that the potential benefits of the test outweighed its limitations.

From the perspective of clients, the opportunity to detect HIV a month earlier than other tests was compelling and the longer time to results (around an hour instead of 15 minutes) was deemed a worth-while trade-off. One client explained, *"An hour's not too bad... it would be an hour of torture, but yeah, that's pretty good… that's impressive. It's worth it… to have that opportunity"* (Terrence, client). Another elaborated on the implications of earlier diagnosis, noting that *"these two or three month periods of time that it takes to find out [if you have HIV]. That's a lot of life going on in that period of time. And if you want to try to let all the people know that you might have come in contact within those two to three months. That's not easy… I think it's [the AHI test] a great idea. For all kinds of reasons, not just that there's less time to be infecting other people, there's an easier chance of finding the one that infected you"* (Terrence, client).

Staff also recognized important benefits. Several pointed out that the test could facilitate earlier initiation of treatment, especially for high-risk individuals who might otherwise be lost to follow-up. As one medical assistant explained, *"…if they tell me it's been three or four weeks ago, since they had the unprotected sex... my biggest worry is: are they really going to come back in 90 days?.... So, yeah, I think that [AHI test] would be great. Even if it took a little bit longer, and I had to hold them here in the clinic a little bit longer, if it tested sooner, that's great, because you're not always going to capture people to come back within 30-90 days"* (Ana, Medical Assistant). Staff also highlighted how the test could reduce anxiety, particularly for those who had experienced high-risk or traumatic events: *"if it's someone who's testing because of the specific experience, especially if it was non-consensual or something like that, they've already had to wait so long when why wait longer than necessary"* (James, HIV and STI Tester).

Despite this enthusiasm, some staff also raised concerns related to feasibility and client expectations. Rapid tests are often marketed on their speed, and one tester acknowledged, "*it's hard though, because with the rapid tests we have now, that's one of our gimmicks, you know, it's like 'it only takes a minute', but I think as long as you work things in a way that make sense for your organization, then it can definitely be worth it*" (James, HIV and STI Tester). Others worried about the operational challenges of longer testing times, particularly in busy clinics: "*It depends… we try to see everyone within 30-45 minutes. So then, [proposed rapid AHI test] will probably not be the best option for our site, but it may be the best option for someone else*" (Morgan, Testing and Counseling Program Manager).

### Target End-users

Possible target end-users discussed for this test were: 1) client (i.e., self-testing), and 2) testing by a CHW or peer recovery specialist (i.e., CHW/peer-based testing). Table 1 outlines the benefits and limitations of each end-user as perceived by participants.

Overall, there was a greater preference for CHW/peer-based testing over self-testing, however clients were more comfortable and open to self-testing than staff. Despite staff hesitancy, most recognized that self-testing has "its place" and could serve as "a good tool for people to have", or at the very least can be a last resort to reaching others who wouldn't otherwise get tested. Participants suggested that conducting the HIV rapid test could be done by anyone who is adequately trained, including on follow-up and linkage to care, "*training doesn't necessarily mean they have to be a doctor or a nurse. It just means that they should be prepared to give the results, whatever the results may be…and if it were to turn out positive, then somebody who knows how to get connected to care*" (Terrence, client). These skills are some of the benefits mentioned for CHW/peer-based testing in Table 1.

Additionally, organizations preferred CHW/peer-based testing, especially for high-risk or hard-to-reach communities like PWID,

*"either peer-led or community health testing… I think would definitely benefit [PWID] … we're able to test quite a few injection drug users just because we're going to where they're at, and we have an education component … nobody wants to be tested in the beginning, but then as the educator is halfway through the education, 6 of the 10 now want to be tested." (Kennedy, Director of Outreach Services)*

### Implementation Considerations

Given the benefits and concerns surrounding both self-testing and CHW/peer-based testing, participants also shared key considerations for implementing an AHI test. For self-testing, **accuracy** was described as a key concern. As one client explained, "*As long as I can trust the accuracy and all that of the test. Yeah. Because, you know, why not get tested [laughs]?*" (Terrence, client). **Cost** was also frequently mentioned, with organizational staff pointing out the contrast between high retail prices and free community testing: "*There's also just a little frustration with the fact that I mean if you purchase one at CVS that are like $50, and you can come to us and get tested for free*" (Evy, Prevention Team Lead). Clients emphasized the importance of clear and **simple instructions** to ensure usability, noting that "*I think as long as the instructions were clear and everything that would, that would be fine to me*" (Kiana, client). Several participants also stressed the importance of providing **guidance for next steps** following a positive result. One client remarked, "*When these type of tests are going to be over the counter… they should also have some information about where people can go to do to do next, if they test positive*" (Jaime, client). Finally, both clients and providers cautioned against the isolation that self-testing might create, underscoring the need for **support systems**. As one outreach director stated, "*I would just hope that there was a human element, a way for someone to reach out to somebody that, just to help them deal with the information they received, you know*" (Blaire, Director of Outreach and Testing). Some organizations who implemented a

**Table 1. Benefits and Concerns regarding different target end-users of the rapid AHI test.**

| Self-testing | Benefits | |
|---|---|---|
| | Increases access/ addresses barriers | "I think **self-testing, it's got its place**. I think if they're not going to come in here, because of the stigma, it's great." – Ana, Medical Assistant<br><br>"I think especially with a lot of people who use IV drugs, it's kind of an isolating lifestyle. And I think it becomes a barrier to healthcare, having to… seek out resources for something… that there's a giant social stigma for, and also, like, sometimes it's just hard to find those programs. **Being able to do it at home would eliminate pretty much all obstacles.**" – Kiana, client |
| | Convenience | "I could do it… on my own schedule as opposed to going to a clinic" (Robert, client) |
| | Privacy* | "I think it'd be cool to be able to do it yourself, because a lot of times, I feel like some people might have that like sliver of doubt, to where, you know, they- if something did happen, maybe they would just want to be able to test themselves in private." – Robert, client |
| | Peace of mind* | "I can be by myself and if I imagine any wonders or issues or whatever, then it's just easy just do it and, you know, go on from there. Like I said, at least I know that I'm good on my end of it" – Natalie, client |
| | **Limitations or Concerns** | |
| | Accuracy* | "… like with pregnancy tests or with any other thing like that, there's always that doubt of, well, you know, maybe it's maybe it's wrong, maybe I'm gonna have to go to a doctor to check and stuff" – Robert, client |
| | User ability to perform test correctly | "… not everyone is great at like following directions… you might get an unclear result if you're not following directions clearly" – Kiana, client |
| | User ability to interpret results** | "I just think that having somebody do a home test, they're just not going to understand. You know, all they're going to see is that that line, or those two lines, and they're going to think they're positive, and then being able to reach somebody. I mean if somebody tests on a Friday night at home, they would not be able to reach say our organizations until the following Monday because we're not open on the weekend – Kennedy, Director of Outreach Services |
| | Self-reporting** | However, my concern is the self-reporting. If I'm self-testing, you're home alone, you know, am I going to report myself if it does come out reactive? – Morgan, Testing and Counseling Program Manager |
| | Lack of pathway to follow-up treatment | I guess the biggest downside would be concerned about what action to take, willingness to take action?" – Jaime, client<br><br>"… self testing is great but would they really let somebody know if they were positive, to be able to get into medical management, you know, treatment in that way. And you have to remember like we work a lot with people who use drugs, who inject. So, some of them might not be very forthcoming with that or just might not ever do anything about it. Sometimes it takes that extra push from us. You know it's not a death sentence anymore, you can get better, we can do treatment, you know some of that encouragement. **I don't know that we would ever get them into care, that's my biggest concern around here."** – Kayla, HIV Prevention Outreach Coordinator/ Certified Peer Recovery Coach |
| | Lack of emotional support and counseling | **"If a person turned out positive, they're kind of all on their own at that moment.** And that would be a traumatic moment… You don't test yourself for cancer and find out all alone that you have cancer. So that would be dramatic." – Terrence, client<br><br>"I think my main concern about that [self-testing] is especially with HIV and just because of those stigmas we talked about earlier, the counseling part after a positive test… there's that support in an office of people [who] are educated in one, crisis intervention, and two, in HIV. I think is very important and if you test at home, you don't have that component of it." – Gia, Harm Reduction Program Manager |
| | Potentially exacerbating stigma | "But I think with advancing the move away from stigma and historical trauma of HIV from you know the 80s and the 90s, and the silencing and some of the public service campaigns that maybe dramatize HIV in not so helpful ways, I think that as we push to really get stigma dealt with, **it's not necessarily helpful to promote the idea that HIV testing is something that a person needs to do at home**… it's not quite that, it's really not quite that sensitive." – Anita, HIV Testing Department Manager |
| CHW/ Peer-based testing | **Benefits** | |
| | Addresses access barriers | "You look at some of our counties we serve, they're very very small, and they have no access to any testing whatsoever and if we didn't go there, on a regular basis and do testing at a treatment center or at the county health department… They wouldn't do it… **you need to meet people where they're at.**" – Kennedy, Director of Outreach Services |
| | Skilled at building trust and rapport** | "… as a community health worker you could get in there and really begin to understand what their needs are and what their wants are, and you could slowly bring in other people to kind of meet that [need]…to where it's not just them coming to you... I think it can be a great way to combat HIV." – Blaire, Director of Outreach and Testing |

*(Continued)*

**Table 1.** (Continued)

| Self-testing | Benefits | |
|---|---|---|
| | Can provide education, guidance and support | *"I think community health workers are helpful and they can help you understand a wide variety of things having to do with your health… somebody who can make sure that you understand what's happening, right?"* – Terrence, client |
| | Unique bonds through shared life experiences | *"… we've had someone that had previous injection drug use and we tested and they were reactive. That same person [tester] was also living with HIV. The instant bond that they had and [how they] were able to communicate about it, their drug of choice that they both used, was an automatic bond… I have years of experience, but the rapport and how quickly they built that rapport with each other… and [to] have a really deep understanding of what the feelings that they were kind of going through even though they were different people, they had those similarities that were, I think it was really beneficial to that person."* – Blaire, Director of Outreach and Testing<br><br>*"it's good when there's someone, like I said, that knows kind of about it [drug use] because I hate people that they get hired at some of these places and then they haven't, they've never lived it, they've never seen it and it's hard to take somebody serious…"* – Natalie, client |
| | Can provide a safe, non-judgmental space** | *"It creates a sense of safety. The judgmental piece goes away, the awkwardness… I'm a person who used to inject drugs and so when I can talk about certain things and just like the way they talk about them and not the way, you know, like a medical person or a clinical person would talk about them. There's, like, almost a sense of just like you can just feel them like loosen up and see them relax"* – Gia, Harm Reduction Program Manager |
| | **Limitations or Concerns** | |
| | Limitations caused by budget and institutional policies** | *"Well, we are limited by our budget and we're limited by what we as an institution can do… we used to go to the jails, and then they changed the rules... so now we can't"* – Andy, Outreach Coordinator and Tester, Community Health Worker |
| | CHW/Peer capacity** | *"Our limitations are also based off of what the worker is capable of doing with their time, like right now in our office there's two of us. … I would say like rural communities are extremely difficult, because, we had a few sites that are like, an hour away. And that's the general limit that we can do because mileage, and how far we're willing to drive on our time"* – Andy, Outreach Coordinator and Tester, Community Health Worker |
| | Gaining community trust is hard work** | *"I think sometimes it's seen as not as effective only simply because the people aren't putting in the work to do it. They're not putting in the work to understand the culture that they're going into, and they become, they get rejected very quickly, and they just don't put the effort in or they don't have an understanding, or they communicate that no one's at that venue. They just haven't done the hard work"* – Blaire, Director of Outreach and Testing |
| | CHW/Peer safety and emotional wellbeing** | *"We also see occasionally somebody come through who hasn't really dealt with their own baggage, so to speak… I've seen I think a lot of projection, I think a lot of countertransference, I've seen a lot of re-traumatization happen in people, if they haven't done that work. So yeah, absolutely peer based testing, hell yeah, but whoever's managing that needs to be conscientious of the ways that that person that that staff member, that peer-based tester, can also get re-hurt, you know, or continually harmed in that process."* – Anita, HIV Testing Department Manager |

*only expressed by clients
**only expressed by staff

mail-delivered self-testing program during the COVID-19 pandemic [31] also shared important considerations for connecting with high-risk clients to meet them where they are.

For CHW and peer-based testing, participants highlighted the importance of adequate **training** to ensure accuracy and reliability. One outreach coordinator explained, "*If we can train them and get them to read the results accurately, then I'm all for it*" (Andy, Outreach Coordinator). Others emphasized that success depended on recruiting individuals with strong **community connections and interpersonal skills**. As a prevention team lead put it, "*…we have to really befriend the communities that we want to get to know and earn their trust… finding those folks who can help build that relationship is really important*" (Evy, Prevention Team Lead). Additionally, some suggested that the AHI test should be **integrated into existing testing menus** to give clients more options. As one HIV/STI tester explained, "*Maybe [this test] can be something that a patient has a choice of [you tell them] 'well this could find it [HIV] in two months versus this one finds within three months and the time periods...'*" (Carissa, HIV and STI Tester).

Together, these perspectives highlight that while both self-testing and CHW/peer-based testing hold promise for expanding access to AHI detection, successful implementation will depend on careful attention to issues of accuracy, affordability, usability, support, training, and trust-building within communities.

## Discussion

This study aimed to understand context-specific challenges to HIV testing in Indiana, assess the acceptability of a novel AHI test in this context, and identify the target end-user and related implementation considerations for the technology. The required waiting period (up to 90 days post-exposure) of existing rapid tests is a significant barrier to preventing transmission and facilitating access to care in these populations. To that end, the need to detect HIV earlier among high-risk groups was expressed by many participants, and a novel AHI rapid test that would allow for this, despite taking up to an hour to produce results, was deemed highly acceptable. These findings both support and inform the continued development of this novel device.

This work can inform the development and implementation of broader emerging AHI tests. Portable RNA tests for POC AHI detection have recently been developed but not yet implemented [32–35]. Additionally, alternative antigen-antibody 4th generation POC tests have been developed to detect AHI at the POC [36], however, both field and multi-laboratory trials have shown that the antigen detecting capabilities of these tests were unsuccessful [37,38]. With additional incubation or sample preparation steps [39] these 4th generation tests may be better able to detect AHI. Our findings demonstrate that these additional steps may be worthwhile despite the added time to result, as our participants found this tradeoff to be acceptable if it meant earlier AHI detection.

While the FDA has approved qualitative laboratory-based HIV RNA tests for diagnosis, current CDC testing algorithms only recommend using laboratory RNA tests after HIV antigen/antibody testing for confirmation or genotyping, or if the first tests are indeterminant [40]. RNA-based AHI tests are more expensive than antibody/antigen counterparts [41]. Babigumira et al compared found that offering POC RNA-based testing to outpatients who present with symptoms of AHI would be a cost-effective strategy [42]. Similar analyses for screening of high-risk individuals in lower-burden countries would also be important to understand potential costs for implementation effectiveness in the US.

To inform further design of this novel test, different testing modalities were discussed with participants to gauge preferences for the target end-user (self-testing vs CHW/peer-led testing) and related implementation considerations. While participants expressed important benefits and concerns for both end-user groups, the majority felt that CHW/peer-led testing would better meet the needs of high-risk populations, particularly regarding the need for counseling and navigation to care. These findings support a growing body of literature on the acceptability of both HIV community-based/peer-led and self-testing more broadly [43–49]. Several reviews have found that while interest in self-testing is high among vulnerable groups because of convenience, privacy, and user desire to keep track of their status [50,51] many are concerned about the reliability and accuracy of tests and linkage to care and counseling [44–46]. These are many of the same concerns that existed when HIV antibody self-tests were first instituted [52]. Clear and simple instructions for self-tests, including next steps for a positive result, are important implementation considerations. With these in mind, it has been well documented that self-testers can reliably and accurately conduct rapid HIV testing compared to trained healthcare providers [53]. Future work will prioritize usability testing of the resulting rapid AHI test for both CHW/peers and clients to ensure ease of use.

This study also contributes to literature on model approaches for appropriate technology development that meaningfully consider and incorporate contextual investigation early in the design process of novel health technologies [20]. The utilization of human-centered design frameworks and engagement of diverse stakeholders to inform the acceptability and preferences of HIV testing interventions are key strategies to understand and promote their successful implementation, uptake and adoption [54,55]. Relatedly, a key limitation of this study is that recruitment relied on flyers posted at HIV service organizations, which likely biased the client sample toward individuals already engaged in HIV testing or care. As a result, the

perspectives of medically underserved or less connected populations may not be fully captured. Future research should adopt proactive recruitment strategies, such as outreach in non-clinical community settings, to reach individuals not currently accessing testing or care. Given the sensitivity of the topic and the vulnerability of this population, our study relied on community partnerships with established client relationships. As such, our sample size of clients was relatively small and thematic saturation was not reached among this participant group. Recruitment was constrained by the ethical need to rely on passive, low-burden strategies during the COVID-19 pandemic. These constraints limited sample size but were necessary to ensure participant safety and confidentiality. Nonetheless, our findings provide important context-specific insights for technology development and offer significant support for future research around AHI diagnostics.

The staff we interviewed represented 10 organizations spanning six counties in Indiana, including health departments, syringe service programs, and healthcare clinics. While this diversity captures a range of service models and settings, it may not fully represent all geographic areas of the state or other organizational perspectives. Our study did not include perspectives from broader stakeholders such as policy makers or payers, who could provide important insights into regulatory and funding considerations. Future work must reach broader stakeholders including local health departments and insurance companies to understand implementation considerations related to reporting and reimbursement. Because the study was conducted in Indiana, findings may not generalize to regions with different HIV prevalence rates, healthcare infrastructure, or sociopolitical environments. Further studies in diverse geographic and epidemiologic contexts are needed to validate and expand these findings.

Given the positive response towards the goal of detecting AHI with POC technology, our findings have informed key next steps in accuracy, simplicity to use, and cost considerations. To increase accuracy, the limit of detection is being improved with new RNA assay designs. Further, the manufacturing costs are being reduced with streamlined device design that uses fewer parts and requires fewer steps for assembly. Lastly, an internal amplification control ensuring test validity and easily interpretable results will be adapted to the AHI detection.

A detailed target product profile (TPP) describing HIV test sensitivity, specificity, cost, failure rates, time-to-result, and operating conditions for at-home HIV testing was previously developed in 2014 [56]. However, this was focused on post seroconversion detection of HIV. While newer TPPs for recent HIV infection have been developed, these focus on epidemiologic surveillance [57], a new TPP with detailed specifications for POC AHI testing is warranted. Awareness of this need has long been known and the MicroRAAD device was originally based off of initial targets for NIH-funded development of technologies for AHI self-testing (PAR-17–471, https://grants.nih.gov/grants/guide/pa-files/PAR-17-471.html). These desired characteristics included acceptability among end-users, rapid test results within 1-hour, detection of 1000 virus particles per mL, specificity for circulating HIV clades, ease-of-use within an integrated sample-to-answer system, simple presentation and interpretation of results, refrigeration-independent storage, minimal electricity dependence, and cost effectiveness for end-users. The early-stage contextual investigation and acceptability study presented herein, combined with responsive technology design iteration and usability testing among the intended users, brings together best practices of human- and equity-centered design.

## Supporting information

**S1 Table. Interview Questionnaires.**
(DOCX)

**S1 File. COnsolidated criteria for REporting Qualitative research (COREQ) Checklist.**
(PDF)

## Author contributions

**Conceptualization:** Natalia M. Rodriguez, Jacqueline C. Linnes.

**Data curation:** Natalia M. Rodriguez, Lara Balian.

**Formal analysis:** Natalia M. Rodriguez, Lara Balian, Ishita Kataki, Cealia Tolliver, Julio Rivera-De Jesus.

**Funding acquisition:** Jacqueline C. Linnes.

**Investigation:** Natalia M. Rodriguez.

**Methodology:** Natalia M. Rodriguez.

**Supervision:** Natalia M. Rodriguez, Jacqueline C. Linnes.

**Writing – original draft:** Lara Balian, Ishita Kataki, Cealia Tolliver, Julio Rivera-De Jesus.

**Writing – review & editing:** Natalia M. Rodriguez, Jacqueline C. Linnes.

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
