## [Decision Letter · Decision Letter 0]

21 Aug 2025

PGPH-D-25-01440

Stakeholder-informed design of a rapid test for detection of acute HIV infection

Dear Dr. Rodriguez,

Thank you for submitting your manuscript to PLOS Global Public Health. After careful consideration, we feel that it has merit but does not fully meet PLOS Global Public Health’s publication criteria as it currently stands. Therefore, we invite you to submit a revised version of the manuscript that addresses the points raised during the review process.

We look forward to receiving your revised manuscript.

Kind regards,

Guillaume Fontaine, PhD, RN

Academic Editor

Journal Requirements:

i. State the initials, alongside each funding source, of each author to receive each grant.

ii. State what role the funders took in the study. If the funders had no role in your study, please state: “The funders had no role in study design, data collection and analysis, decision to publish, or preparation of the manuscript.”

2. Please send a completed 'Competing Interests' statement, including any COIs declared by your co-authors. If you have no competing interests to declare, please state "The authors have declared that no competing interests exist". Otherwise please declare all competing interests beginning with the statement "I have read the journal's policy and the authors of this manuscript have the following competing interests:"

3. Please ensure that your Ethics Statement is available in its entirety at the beginning of your Methods section, under a subheading 'Ethics Statement'. It must include:

1) The name(s) of the Institutional Review Board(s) or Ethics Committee(s)

2) The approval number(s), or a statement that approval was granted by the named board(s) 

3) (for human participants/donors) - A statement that formal consent was obtained (must state whether verbal/written) OR the reason consent was not obtained (e.g. anonymity). 

NOTE: If child participants, the statement must declare that formal consent was obtained from the parent/guardian.

4. Please upload separate figure files in .tif or .eps format. Also, remove the figures from your manuscript file but keep the legends.

5. We have noticed that you have uploaded Supporting Information files, but you have not included a list of legends. Please add a full list of legends for your Supporting Information files after the references list.

6. In the online submission form, you indicated that “Data, coding schemes, and interview guides are available by request. Please email Natalia Rodriguez at natalia@purdue.edu.”. 

3. Uploaded as supplementary information.

7. Some material included in your submission may be copyrighted. According to PLOS’s copyright policy, authors who use figures or other material (e.g., graphics, clipart, maps) from another author or copyright holder must demonstrate or obtain permission to publish this material under the Creative Commons Attribution 4.0 International (CC BY 4.0) License used by PLOS journals. Please closely review the details of PLOS’s copyright requirements here: PLOS Licenses and Copyright. If you need to request permissions from a copyright holder, you may use PLOS's Copyright Content Permission form.

Potential Copyright Issues:

Figure 1: Please confirm whether you drew the images / clip-art within the figure panels by hand. If you did not draw the images, please provide (a) a link to the source of the images or icons and their license / terms of use; or (b) written permission from the copyright holder to publish the images or icons under our CC-BY 4.0 license. Alternatively, you may replace the images with open source alternatives. See these open source resources you may use to replace images / clip-art:

- https://openclipart.org/

Additional Editor Comments (if provided):

Thank you for submitting your manuscript to PLOS Global Public Health.

In addition to the reviewers' comments, please address the following points:

1. The title of the article does not reflect the objectives or the design of the study; please adjust.

2. Abstract: The results section is underdeveloped; please expand and condense the Background.

3. Methods: Add a short paragraph on study design, cite the COREQ guidelines and attach these as an appendix, making sure all of the items in the checklist are presented in the text.

4. Methods: Please clarify what types of organizations were invited to participate (HIV organizations is somewhat vague).

5. Results: Please nuance your statement "Acceptability of the described AHI rapid test was high." - You then state "All participants agreed there was a need to detect HIV earlier in high-risk populations." however this does not directly support your first statement. There is a need to significantly expand what is presented in this section, differentiate it by clients and staff which probably have different perspectives of acceptability, and beyond saying that acceptability is high you should describe the reasons for this. Consequently, I suggest transforming table 1 into narrative text to expand the acceptability section.

6. Results: Table 3 does not add value as it is; please transform it to narrative text to expand the section "Implementation Considerations."

Finally, please clarify whether the senior author holds any commercial interest in the test under evaluation, and expand on the safeguards implemented to ensure the integrity and impartiality of the study findings.

Academic Editor Guillaume Fontaine

Reviewers' comments:

Reviewer's Responses to Questions

**Comments to the Author**

1. Does this manuscript meet PLOS Global Public Health’s publication criteria?

Reviewer #1: Yes

Reviewer #2: Yes

Reviewer #3: No

2. Has the statistical analysis been performed appropriately and rigorously?

Reviewer #1: N/A

Reviewer #2: N/A

Reviewer #3: No

3. Have the authors made all data underlying the findings in their manuscript fully available (please refer to the Data Availability Statement at the start of the manuscript PDF file)?

Reviewer #1: No

Reviewer #2: Yes

Reviewer #3: Yes

4. Is the manuscript presented in an intelligible fashion and written in standard English?

Reviewer #1: Yes

Reviewer #2: Yes

Reviewer #3: Yes

Reviewer #1: This is a well-written and timely qualitative study that explores stakeholder perspectives on a novel rapid test for acute HIV infection. The manuscript is grounded in human-centered design and presents meaningful insights for public health implementation, especially among high-risk populations. The methods are appropriate, the findings are clearly presented, and the discussion is thoughtful. With minor revisions to improve clarity in the abstract and introduction, the manuscript will make a strong contribution to the literature on HIV diagnostics and equity-focused health innovation. However, there are space for improvements.

1. Abstract: Clarify the purpose and novelty of the AHI test earlier, and state the main finding more directly (e.g., "Participants supported earlier detection despite longer wait times").

2. Introduction: Briefly introduce the AHI test earlier and focus less on technical mechanisms; emphasize its public health significance and relevance for high-risk populations.

Reviewer #2: This is an important area of research that may reduce the transmission of HIV

The manuscript is well-written, although it contains a few grammatical errors that need correction.

1 The abstract does not have the aim of the study.

2. Why did the authors like our other stakeholders who made meaningful contributions to the study? These may include opinion leaders, Ministry of Health decision makers, community representatives, etc

3 According to the authors, do they think the stakeholders represented the various sectors and geographical locations?

Reviewer #3: I acknowledge the efforts of the authors in examining the acceptability and implementation considerations for a novel rapid test designed to detect acute HIV infection (AHI). The study addresses a critical gap in HIV testing by focusing on AHI detection, a period of high transmissibility. It aligns with the "Ending the HIV Epidemic" initiative. Unfortunately, several limitations are noted demanding major revisions.

Some of these are highlighted below;

• The claim that "none can accurately detect HIV in the early AHI phase" is too broad and doesn't reflect the capabilities of lab-based NAT and fourth-generation assays. The authors should clarify that they are primarily discussing the limitations of rapid, point-of-care tests, not all HIV tests. The importance of rapid AHI tests lies in making early detection more accessible, particularly to high-risk populations who may not regularly visit clinics or have easy access to lab testing.

• The small sample size of clients (n=5) raises concerns about the generalizability of their views. The authors acknowledge that thematic saturation was not reached among this group.

While the authors mention the challenges in recruiting clients, further efforts to increase the sample size or a more in-depth justification for the limited sample would strengthen the study.

• Recruiting clients via flyers posted at HIV organizations may have biased the sample towards those already engaged in testing services.

The authors research should consider more proactive recruitment strategies to reach medically underserved communities not currently engaged in HIV testing or care.

• More detailed demographic information about the staff and client participants should be included. While the client ethnicities are mentioned, the economic background of the clients and the number of years of experience of the staff would be relevant.

Adding this information would help to better understand the context of the data.

• The study is based on data from Indiana, which may limit the generalizability of the findings to other geographic locations or populations with different HIV prevalence rates or healthcare systems.

The authors should acknowledge this limitation and encourage future research in other settings.

• The authors mention the need for a new TPP with detailed specifications for POC AHI testing but do not provide a specific TPP for the MicroRAAD test.

Elaborating on the desired characteristics of the AHI test, such as sensitivity, specificity, and cost, would be beneficial.

• Inconsistencies: Line 47, it says "those who test positive to quickly engage in HIV treatment and reduce viral load to untransmissible levels". However, line 56 says, "These tests require a waiting period of up to 90 days after exposure as they cannot detect the virus at the AHI stage". These statements are in conflict with each other and should be addressed in the paper.

**Do you want your identity to be public for this peer review?** For information about this choice, including consent withdrawal, please see our Privacy Policy

Reviewer #1: No

Reviewer #2: **Yes: ** Ester Acen

Reviewer #3: No

---

## [Editor Report · Decision Letter 1]

8 Oct 2025

Stakeholder considerations on acceptability and implementation of a novel rapid test for acute HIV infection: a qualitative study in Indiana

PGPH-D-25-01440R1

Dear Dr. Rodriguez,

We are pleased to inform you that your manuscript 'Stakeholder considerations on acceptability and implementation of a novel rapid test for acute HIV infection: a qualitative study in Indiana' has been provisionally accepted for publication in PLOS Global Public Health.

Best regards,

Guillaume Fontaine, PhD, RN

Academic Editor